# Vorinostat Improves Myotonic Dystrophy Type 1 Splicing Abnormalities in DM1 Muscle Cell Lines and Skeletal Muscle from a DM1 Mouse Model

**DOI:** 10.3390/ijms24043794

**Published:** 2023-02-14

**Authors:** Nafisa Neault, Aymeric Ravel-Chapuis, Stephen D. Baird, John A. Lunde, Mathieu Poirier, Emiliyan Staykov, Julio Plaza-Diaz, Gerardo Medina, Francisco Abadía-Molina, Bernard J. Jasmin, Alex E. MacKenzie

**Affiliations:** 1Children’s Hospital of Eastern Ontario Research Institute, Ottawa, ON K1H 5B2, Canada; 2Department of Cellular and Molecular Medicine, Faculty of Medicine, University of Ottawa, Ottawa, ON K1H 8M5, Canada; 3Eric Poulin Center for Neuromuscular Disease, Faculty of Medicine, University of Ottawa, Ottawa, ON K1H 8M5, Canada; 4School of Pharmaceutical Sciences, Faculty of Medicine, University of Ottawa, Ottawa, ON K1H 8M5, Canada; 5Department of Biochemistry and Molecular Biology II, School of Pharmacy, University of Granada, 18071 Granada, Spain; 6Instituto de Investigación Biosanitaria IBS.GRANADA, Complejo Hospitalario Universitario de Granada, 18014 Granada, Spain; 7Institute of Nutrition and Food Technology “José Mataix”, Biomedical Research Center, University of Granada, Armilla, 18016 Granada, Spain; 8Department of Cell Biology, School of Sciences, University of Granada, 18071 Granada, Spain

**Keywords:** myotonic dystrophy type 1, rare diseases, vorinostat, new treatment

## Abstract

Myotonic dystrophy type 1 (DM1), the most common form of adult muscular dystrophy, is caused by an abnormal expansion of CTG repeats in the 3′ untranslated region of the dystrophia myotonica protein kinase (DMPK) gene. The expanded repeats of the DMPK mRNA form hairpin structures in vitro, which cause misregulation and/or sequestration of proteins including the splicing regulator muscleblind-like 1 (MBNL1). In turn, misregulation and sequestration of such proteins result in the aberrant alternative splicing of diverse mRNAs and underlie, at least in part, DM1 pathogenesis. It has been previously shown that disaggregating RNA foci repletes free MBNL1, rescues DM1 spliceopathy, and alleviates associated symptoms such as myotonia. Using an FDA-approved drug library, we have screened for a reduction of CUG foci in patient muscle cells and identified the HDAC inhibitor, vorinostat, as an inhibitor of foci formation; SERCA1 (sarcoplasmic/endoplasmic reticulum Ca^2+^-ATPase) spliceopathy was also improved by vorinostat treatment. Vorinostat treatment in a mouse model of DM1 (human skeletal actin–long repeat; HSA^LR^) improved several spliceopathies, reduced muscle central nucleation, and restored chloride channel levels at the sarcolemma. Our in vitro and in vivo evidence showing amelioration of several DM1 disease markers marks vorinostat as a promising novel DM1 therapy.

## 1. Introduction

Myotonic dystrophy type 1 (DM1) is the most common form of adult muscular dystrophy with a worldwide prevalence of 1 in 8000 [1]. A more recent assessment of the prevalence of dystrophia myotonica protein kinase (DMPK) CTG expansion repeats in newborns (higher than 50 CTG repeats) suggests this to be an underestimate and the actual prevalence lies closer to 1 in 2100 [2]. DM1 is a multisystemic disorder characterized by a diverse array of signs and symptoms which affect most organs. Signs and symptoms typically include myotonia (sustained muscle contraction), muscle wasting, localized muscle weakness, cardiac conduction defects, testicular failure, cataracts, and insulin resistance, all of which contribute to a significantly diminished quality of life for DM1 patients [3,4,5,6,7,8]. In addition to significant morbidity, there is also DM1-associated mortality. A 10-year longitudinal study of a 367 patient-cohort identified respiratory insufficiency and cardiac failure as the leading causes of death in patients with DM1 [9]. The socio-economic impact of DM1 is significant, as job retention is difficult and compounded by regular visits to different specialists needed for treating the array of symptoms. Accordingly, a therapeutic agent targeting the molecular defect underlying DM1, and thus with the most potential to address the condition as a whole in the various affected systems would be a welcome advance.

DM1 is an autosomal dominant trinucleotide repeat expansion disorder (TRED) which is caused by extended CTG tracts in the 3′ untranslated region (UTR) of the *DMPK* gene [10,11,12,13]. The greater the number of CTG repeats, the more severe the symptoms and the earlier the age of onset [14]. The most widely accepted DM1 pathogenic model invokes RNA gain-of-function with pathogenically expanded *DMPK* mRNA forming hairpin structures through C-G base-pairing [15,16], which aggregate into nuclear foci and cause misregulation and/or sequestration of a number of RNA-binding proteins (RBP) [17]. Chief among these RBPs is the transcriptional regulator muscleblind-like 1 splicing regulator 1 (MBNL1), which binds to the (U/C)GC(U/C) nucleotide sequence and thus has a direct affinity for the CUG repeats in the DMPK foci (CUGCUG) [18,19,20]. *MBNL1* is essential for mRNA alternative splicing, and its sequestration to CUG foci (and depletion from the nucleoplasm) consequently results in widespread aberrant splicing events. Accordingly, DM1 has been termed a spliceopathy [21]. DM1 spliceopathies culminate in the expression of fetal forms of mRNAs not normally expressed postnatally, which in turn leads to many of the DM1 symptoms. MBNL1 has therefore been shown to be central to DM1 pathogenesis and thus represents an attractive therapeutic target. For example, knock-down of MBNL1 in unaffected mice results in a DM1-like phenotype [22], while exogenous *MBNL1* overexpression in DM1 mice, likely exceeding the sequestration capacity of the foci, corrects splicing defects and related symptoms [23].

It has been previously shown that disaggregating these RNA foci by antisense oligonucleotide (ASO) or morpholino repletes MBNL1 [24,25,26], rescues the DM1 spliceopathy, and alleviates signs and symptoms including myotonia [24,25]. Recent work shows promise in this realm [27] in muscle [28] and brain tissue [29].

An alternative to molecular therapies is the use of small-molecule compounds to regulate key DM1 pathways. Given the central role of CUG RNA foci in DM1 pathogenesis and the fact that the dissolution of foci (e.g., ASO treatment) reverses DM1 in mouse models [25], we have used intranuclear CUG RNA foci integrity as a therapeutic target. We present here the results of a high-throughput screen using FDA-approved drugs that identified small molecules with the capacity to reduce CUG foci formation in patient cells, and ultimately correct DM1 spliceopathy and associated signs in vivo. Such a repurposing approach could lead to the rapid development and implementation of novel clinical interventions with significant efficacy for DM1.

## 2. Results

### 2.1. Immortalized DM1 Patient Myoblasts Display Hallmark Features of Disease: CUG RNA Foci and Aberrant Splicing

Two immortalized DM1 patient myoblast cell lines and three immortalized control myoblast cell lines were acquired through the Euro Biobank, made available by Pantic et al. Although originally reported to have greater than 500 CTG repeats [30], our own Southern blot analysis demonstrated that these cells contained approximately 3600 (DM1_3600CTG_) and 3300 (DM1_3300CTG_) CTG repeats (Appendix A). RNA FISH using an Alexa555-CAG_10_ probe showed that only DM1 patient myoblasts had detectable nuclear CUG-foci, which were absent in control myoblasts examined under a microscope (Appendix A). In addition, differentiated DM1 cells had approximately three times more foci than proliferating DM1 myoblasts (Appendix A). This was in line with the increased expression of DMPK mRNAs in differentiated versus proliferative myoblasts, as measured by RT-qPCR (Appendix A). Proliferative DM1 myoblasts also had dysregulation of SERCA1 splicing (as measured by RT-qPCR), a classic feature of DM1 [31,32], which was more pronounced in differentiated myoblasts (Appendix A).

### 2.2. High Throughput Screening of FDA-Approved Compounds Identified Small Molecule Candidates for CUG Foci Reduction

Due to the higher number of foci observed with DM1_3600CTG_ cells, it was selected over DM1_3300CTG_ cells for screening. Compared to control ASO, there was a 30% reduction in CUG-foci with 20 nM DMPK ASO treatment and no change in nuclei count (Appendix A). The Z-prime factor (Z’ factor), which reflects the degree of separation between positive and negative controls, can be used to validate a screening assay [33]. Typically, a Z’-factor between 0–0.5 is acceptable, between 0.5–1 is excellent, and 1 is ideal. In this regard, the Z’-factor for the screen parameter, foci area per nuclear area, was 0.84 when comparing the DMPK ASO and control ASO/vehicle treated cells, indicating an “excellent” assay (Figure 1).

The impact of all of the drugs screened on foci is shown in Appendix A. The top three hits, bortezomib, vorinostat, and gemcitabine (Figure 1), were next tested in both DM1 myoblast cell lines, looking at a 100-fold range of drug concentrations for bortezomib (0.1–10 µM) and a 1000-fold range for vorinostat and gemcitabine (0.01–10 µM). Bortezomib showed only mild foci diminution in just one of the cell lines and was cytotoxic at higher doses (Figure 2A,B). Treatment with vorinostat resulted in a dose-dependent decrease in foci area per nuclear area in both DM1_3600CTG_ and DM1_3300CTG_ differentiated myoblasts, starting at 0.5 µM (Figure 2C). This reduction increased up to 10 µM, at which point toxicity, as indicated by the lower nuclei count, supervened (Figure 2D). Similar to vorinostat, gemcitabine also reduced foci in a dose-dependent manner (Figure 2E) and was relatively non-toxic (Figure 2F).

### 2.3. Vorinostat, but Not Gemcitabine, Rescues SERCA1 Spliceopathy

Vorinostat was found to reduce DMPK mRNA levels, as measured by RT-qPCR (Figure 3A). This effect was not seen with a low dose of 0.1 µM, which also did not reduce foci formation (Figure 2C). The reduction in DMPK mRNAs was observed at both a non-toxic concentration of 1 µM and a high/toxic concentration of 10 µM. These data suggest that vorinostat might inhibit foci formation by decreasing *DMPK* mRNA expression. SERCA1 splicing was also assayed by RT-qPCR by measuring the relative SERCA1-A levels compared to total SERCA1-A and B. We noted a vorinostat mediated dose-dependent increase of the proportion of *SERCA1-A* (Figure 3B). Gemcitabine, in contrast, did not reduce *DMPK* mRNA expression (Figure 3C) and failed to affect SERCA1 splicing (Figure 3D). Therefore, gemcitabine was not pursued further.

Zhang et al. documented that vorinostat mediated upregulation of MBNL1 protein in a flow-cytometry-based screen assessing GFP-tagged MBNL1 expression in both artificial HeLa cell models of expanded CTG repeats and DM1 fibroblasts [34]. With this in mind, we assessed MBNL1 protein content in differentiated DM1 myoblasts (Figure 4). Vorinostat increased MBNL1 protein levels in DM1_3600CTG_ myoblasts by up to 2.5-fold but only minimally affected MBNL1 expression in DM1_3300CTG_ myoblasts. In DM1_3600CTG_ myoblasts, there was no change in MBNL1 protein with a 0.1 µM dose, an approximately 1.5-fold increase with a 1 µM dose, and an approximately 2.5-fold increase with a 10 µM dose. The induction in MBNL1 protein expression as well as greater bioavailability due to reduced foci formation may contribute to the observed normalization of SERCA1 spliceopathy.

### 2.4. Other pan-HDAC Inhibitors Also Alleviate DM1 Pathogenic Features, Similar to Vorinostat

In order to assess whether the vorinostat-mediated normalization of DM1 cellular phenotypes is a result of HDAC inhibition or off-target effects, we assessed two other pan-HDAC inhibitors that function similarly to vorinostat. Belinostat decreased foci formation in both cell lines in a dose-dependent manner, similar to vorinostat (Figure 5A,B). Trichostatin A (TSA) reduced foci formation, although with a higher-degree of variability and had the highest cellular toxicity at lower doses (Figure 5C,D). Together, this suggests a role for HDAC inhibition in the reduction of foci. These additional HDAC inhibitors also reduced *DMPK* mRNA levels and corrected *SERCA1* splicing (Figure 5E,F). *DMPK* mRNAs were also decreased in control myoblasts by all three inhibitors, suggesting a mechanism that is independent of pathogenic CUG expansions (Appendix A).

### 2.5. Vorinostat Showed Promising Therapeutic Effects in the DM1 HSA^LR^ Mouse Model

The impact of vorinostat on a DM1 mouse model overexpressing 250 CTG repeats, driven by the human skeletal actin promoter (HSA^LR^), was next assessed; a colony of wild-type mice (WT) and mice expressing HSA containing non-pathogenic CTG repeat lengths (less than 50) (HSA^SR^) were used as non-diseased baseline models. This well-accepted DM1 mouse model has been widely used in pre-clinical testing of therapeutic candidates for ASO therapy [24], along with small molecules including AICAR (5-aminoimidazole-4-carboxamide-1-β-D-ribofuranoside) [35] and tideglusib [36]. All experiments and analyses were done by individuals blinded to the treatment regimen in keeping with in vivo study recommendations [37]. Initially, a short-term one-week trial was performed in HSA^LR^ mice with daily intraperitoneal (i.p.) injections of vehicle or 25 mg/kg vorinostat. The mice were sacrificed and skeletal muscle tissues from one hind leg was frozen in OCT for sectioning and imaging. In contrast, tissues from the other hind leg were flash frozen in liquid nitrogen and stored for RNA analyses. H&E histological analysis of central nucleation, a marker of regeneration in the face of putative muscle damage, showed an increase in HSA^LR^ mice and was slightly reduced in the treated mice (Appendix A) as well as a trend to normalize RYR1 and SERCA1 splicing as shown by RT-PCR products resolved on polyacrylamide gel (Appendix A).

In an effort to elicit a more profound impact on the DM1 phenotype, animals were given daily i.p. injections of 50 mg/kg of vorinostat for 4 weeks. These injections were increased from 25 mg/kg for 1 week, and then processed as described above. Even at a higher dose for longer periods, the drug was well tolerated with no overt weight loss or other observable toxic effects (Appendix A).

RT-qPCR analyses revealed that a 4-week course of vorinostat resulted in significant rescue of dysregulated RYR1 splicing, which has been linked to muscle weakness [30], as well as normalizing SERCA1 splicing in TA muscle (Figure 6A). In *extensor digitorum longus* (EDL), RYR1 and SERCA1 splicing correction was more modest (Figure 6B). Analysis of chloride channel (CLCN1) splicing, another gene misspliced in DM1 which has been implicated in chloride channelopathy and myotonia [38,39], revealed no dysregulation in the *tibialis anterior* (TA) from untreated HSA^LR^ mice compared with that from wild-type mice (Figure 6A). However, significant dysregulation of CLCN1 splicing was observed in HSA^LR^ mouse EDL; this was corrected with vorinostat treatment. (Figure 6B). RT-qPCR analysis of the transgene showed that vorinostat reduced HSA transcript levels in TA muscle (Appendix A).

The degree of central nucleation increase in HSA^LR^ mice was higher in EDL than TA; vorinostat treatment resulted in significant normalization in TA but not in EDL (Figure 7). Fiber size distribution was also quantified using laminin immunofluorescence in EDL (Figure 8 and Appendix A). A higher percentage of larger fibers reflecting mild hypertrophy was observed in HSA^LR^ mice when compared with WT mice; this distribution was not impacted by vorinostat (Appendix A). Finally, histological localization of chloride channels in EDL muscle revealed that, in addition to correcting CLCN1 splicing (Figure 6B), vorinostat also increased the number of CLCN1-positive fibers in DM1 mouse muscles relative to the number of laminin-stained fibers. (Figure 8). On average, 97% CLCN1-positive fibers were observed in muscle from wild-type/HSA^SR^ mice while only 66% positive were observed in the vehicle-treated cohort, increasing to 80% with vorinostat treatment. Collectively, we believe our data point to the potential therapeutic utility of vorinostat for DM1.

## 3. Discussion

Our small molecule screen for modulators of foci integrity in DM1 led to the identification of vorinostat as an agent that inhibits foci formation and modulates DM1 aberrant splicing both in vitro and in vivo. The exact mechanism by which foci reduction occurs is unclear. Similar to DMPK ASO therapy, vorinostat and other HDACi molecules which reduced both foci and splicing dysregulation also decreased *DMPK* mRNA. This suggests that transcript reduction might be the central mechanism of foci reduction. In keeping with this model and as discussed below, despite apparent foci disaggregation, gemcitabine neither decreased *DMPK* mRNA nor improved splicing patterns. Whether vorinostat reduces *DMPK* mRNAs by modulating transcription or transcript stability, or both, is unknown. Even greater reduction in *DMPK* mRNA was documented in vorinostat-treated control myoblast cultures (Appendix A), as compared with the DM1 cell line; it may be that the documented increased half-life of DM1 expanded *DMPK* mRNA [17] blunts the normally brisk vorinostat-mediated downregulation of *DMPK* mRNA levels. The vorinostat mediated reduction in *DMPK* mRNA level does not appear to be linked to the presence of the DM1 mutation as it was observed in both DM1 and control cells. Furthermore, the effect does not seem to be specific to the *DMPK* gene as the HSA^LR^ mice, with 250 CTG repeats driven by the human skeletal actin promoter, also show a reduction in *has* mRNA. A similar impact of vorinostat has*HSA* has been recently documented [40]. Finally, in addition to the increased MBNL1 bioavailability resulting from reduced foci sequestration, the direct induction of MBNL1 protein expression mediated by vorinostat (Figure 4) may also contribute to the observed normalization of DM1 splicing patterns.

Vorinostat, also known as SAHA or commercially known as Zolinza, is a pan-HDAC inhibitor, targeting HDAC 1, 2, 3 and 6. Its use, which increases acetylated histones and alters gene expression, can result in the downstream induction of apoptosis and cell growth arrest [41]. Originally developed as an anti-cancer agent, vorinostat is clinically approved for cutaneous T-cell lymphoma [42] and has been shown to reduce tumor growth and formation in pre-clinical models of metastatic bone cancer, uterine sarcoma, and estrogen receptor-negative mammary tumors [43,44,45].

In addition, there are a number of clinical trials assessing vorinostat for chronic disorders unrelated to cancer. One study explored the tolerability of oral vorinostat (200 and 400 mg, 3 days on/4 days off) for the neurodegenerative lysosomal storage disorder Niemann-Pick disease type C (NPC) (NCT02124083). A second study is determining the maximal tolerable oral daily dose (100 to 400 mg) of vorinostat in Alzheimer’s disease (NCT03056495). A third trial is assessing the safety, tolerability, and efficacy of vorinostat (230 mg/m^2^/day orally for 6 weeks) in children with refractory epilepsy (NCT03894826). Several other clinical trials are assessing the impact of vorinostat alone (NCT01365065) or in combination (NCT03212989, NCT01319383, NCT02475915, NCT03803605, NCT03382834) on human immunodeficiency virus (HIV).

Importantly, the dose of 400 mg used in the NPC clinical trial (NCT02124083) resulted in no serious adverse effects and is associated with peak serum values ranging from 1.8 μM [46] to 2.4 μM [47] within a few hours of oral administration. In this regard, reduction of foci formation (Figure 2C,D), reduction of *DMPK* mRNA, and correction of splicing (Figure 3A,B) all begin to be observed in DM1 myoblast systems at 1 μM.

The potential for vorinostat in treating DM1 was further validated in vivo in HSA^LR^ mice where, in addition to splicing correction, chloride channels levels were restored at the sarcolemma in EDL muscles (Figure 8). The reduction in chloride channel in DM1 has been directly linked to myotonia and chloride channel protein replacement alleviates myotonia [25,39]. One might anticipate, given the broad bioavailability of vorinostat [48], that DM1 affected organs and tissues in addition to skeletal muscle may be beneficially impacted by vorinostat. Unfortunately, this analysis is not possible in HSA^LR^ mice which express the pathogenic transgene in a skeletal muscle-specific manner.

There is limited data on vorinostat pharmacokinetics in mice at what is a comparatively low 50 mg/kg i.p. dose, but mice given i.p. injections of 150 mg/kg attained a plasma level of 200 µM which dropped to approximately 5 µM within one hour [49]. Allometric dose scaling analysis puts the human dose equivalent to the 50 mg/kg used in mice at a clinically attainable 400 mg/day as outlined above [50].

To further explore the role of HDAC inhibition in the observed effects, two additional pan-HDAC inhibitors, belinostat and TSA, were tested in vitro. Both compounds demonstrated a similar effect on foci reduction (Figure 5A,C) and splicing correction (Figure 5E,F) as vorinostat, supporting an underlying class effect involving HDAC inhibition. Interestingly, HDAC3 has been found to promote CTG·CAG expansions in human astrocytes and siRNA mediated knockdown or chemical inhibition of HDAC3 reduced the frequency of expansion events [51]. Whether this was through direct interaction with the repeat region or by an indirect mechanism was not reported. It should be noted that while HDAC3 inhibition and knockdown reduced the frequency of expansion, there was no report of any impact on foci nor on repeat contraction. Such a contraction at the DNA level could be associated with the reduction in CUG foci and correction of downstream splicing (but not the observed reduction in *DMPK* mRNA) by vorinostat and similar pan-HDAC inhibitors. As such, a link between the effects of HDACs on somatic repeat expansion at the DNA level and the reduction in overall *DMPK* mRNA shown in this study remains unclear.

The chemotherapeutic agent gemcitabine was also shown to reduce CUG foci in two DM1 myoblast cell lines (Figure 2). However, unlike vorinostat, gemcitabine had no impact on *DMPK* mRNA levels or *SERCA1* spliceopathy (Figure 3), notwithstanding foci disaggregation. In fact, a trend towards higher *DMPK* mRNA can be seen with increasing gemcitabine concentrations (Figure 3C). It may be that, although microscopically apparent foci are reduced, sub-microscopic *MBNL1* binding hairpin *DMPK* mRNA structures persist in cells treated with gemcitabine. Alternatively, gemcitabine as a cytidine analog [52] mimicking the structure of the nucleotide base cytosine, may bind to the G-rich foci, thereby blocking the binding of the Alexa555-(CAG)_10_ probe, thus masking the foci. Gemcitabine was identified as a DMPK RNA foci inhibitor in a previous screen of DM1 fibroblasts with the effect validated in immortalized and MyoD-converted to myoblasts, but was not further investigated given evidence of cytotoxicity [53]. More recently, foci comprised of GGCCUG repeats in the yeast ortholog of the spinocerebellar ataxia type 36 causing NOP56 gene [54] have also been shown to be disaggregated by gemcitabine, although reduced viability was observed [55].

Although preclinical evidence of therapeutic effects should always be viewed realistically, we believe that the demonstration of meaningful attenuation of DM1 phenotype HSA^LR^ with clinically attainable doses of vorinostat would warrant further exploration of vorinostat either as a monotherapy or in combination with other promising compounds such as tideglusib [36] or metformin [56,57].

Several DM1 disease markers were ameliorated by vorinostat in both in vitro and in vivo experiments. This indicates that vorinostat is a promising novel treatment for DM1 in muscle cells and skeletal muscle. There is a need for further investigation of the mechanism of action of vorinostat in DM1 in other organs, such as the brain and heart.

## 4. Materials and Methods

### 4.1. Cell Culture

All immortalized human DM1 and control myoblasts were obtained as a gift from Dr. Elena Pegoraro’s group [30]. Cells were grown on ECM gel-coated tissue culture plates (corning) in complete myoblast growth media made up of Ham’s F14 Media (VWR, Mississauga, ON, Canada), supplemented with 30% FCS (HyClone, Mississauga, ON, Canada), 10 μg/mL insulin (Sigma, Oakville, ON, Canada), 25 ng/mL bFGF (basic fibroblast growth factor; Life Technologies, Burlington, ON, Canada), 10 ng/mL EGF (epidermal growth factor; LifeTechnologies, Burlington, ON, Canada), 100 units/mL penicillin/100µg/mL streptomycin (HyClone, Mississauga, ON, Canada), and 2 μg/mL Amphotericin B (Gibco, Burlington, ON, Canada) and maintained as an adherent monolayer in humidified incubators at 37 °C and 5% CO_2_. To coat plates with the ECM Gel from Engelbreth-Holm-Swarm murine sarcoma (Sigma, Oakville, ON, Canada), the concentrate was first diluted one-tenth in the base Ham’s F14 media, spread to cover the bottom of the plate, and excess ECM gel recovered for reuse; the ECM gel-coated plate was then incubated at 37 °C for at least 15 min in order to allow the ECM to polymerize on the plate surface.

Cells were passaged before they reached 50% confluence by trypsinization using 1× (0.25%) trypsin (Gibco; reconstituted in 15 mM NaCl (Fisher Scientific, Waltham, MA, USA), 0.5 mM EDTA (Sigma, Oakville, Canada), and 1× PBS (Fisher Scientific, Waltham, MA, USA) for 3–7 min at 37 °C, the trypsin was inhibited by adding complete growth media containing FCS and split at a ratio of one-fifth to achieve 10% seeding density on ECM gel-coated plates. Figure 1, Figure 2 and Figure 5, for foci assay in 384-well plates, myoblasts were seeded at approximately 2000–3000 cells per well (final media volume ranged from 30–40 µL per well); once a confluence of 70–90% was reached, the cells were serum-starved to induce differentiation.

### 4.2. Southern Blot

Southern blot analysis of DM1 myoblasts was performed by the CHEO genetics clinic. Genomic DNA was extracted using DNA mini kit (Qiagen, Toronto, ON, Canada), digested with CutSmart EcoRI-HF (New England BioLabs, Ipswich, MA, USA) for 5 h to overnight, and resolved on 0.6% agarose gel alongside DNA from an individual known to be void of the DM1 genotype (negative control) and DNA from individuals known to have DM1 with pre-established repeat sizes (positive controls); these negative and positive control samples served as representative controls used for diagnostic purposes at the CHEO genetics clinic. The gel was stained with 0.6 µg/mL ethidium bromide in 0.5× TBE for 30 min at room temperature (RT) and imaged. Subsequently, the gel was denatured in a solution of 0.4 M NaOH and 1.5 M NaCl for 210 min under agitation and transferred to Biodyne B nylon membrane (Thermo Scientific, Waltham, MA, USA) in 20× saline sodium citrate (SSC) buffer overnight at RT. Post-transfer, the membrane was neutralized in 0.2 M Tris pH 7.5 and 2× SSC buffer for 15 min at RT, baked between Whatman 3 M paper for 2 h at 80 °C, and pre-hybridized in a PEG hybridization solution of 10% PEG Glycol 8000, 1.5× SSPE, and 7% SDS for 1 h at 65 °C under rotation. Hybridization probes were radioactively labeled with 32P using Prime-It II Random Primer Labeling Kit (Agilent, Santa Clara, CA, USA), purified using Sephadex G-50 Nick column (GE Lifesciences, Chicago, IL, USA), and verified to have radioactivity of 0.3 × 10 mR/h or more per membrane. The radioactive probe was then mixed with 2 mg/mL salmon sperm DNA. It was denatured at 100 °C for 5 min, diluted approximately 1/10 in PEG hybridization buffer, and used to hybridize the membrane at 65 °C for 3 h to overnight under rotation. Following hybridization, the membrane was washed in a solution of 0.1× SSC and 0.1% SDS for 2 × 15 min at RT under agitation. Hybridized membranes were imaged using X-ray film.

### 4.3. High Throughput Screening of FDA-Approved Compounds Identified Small Molecule Candidates for CUG Foci Reduction

All compounds in the FDA-approved small-molecule screen were assayed at a final concentration of 2 µM. DM1_3600CTG_ myoblasts were differentiated for 7 days, treated for 24 h in triplicate plates and fixed in 4% PFA. DNA was stained with Hoechst and CUG RNA foci were detected with an Alexa555-(CAG)_10_ oligonucleotide probe. Cells were imaged using the Opera high content imaging system and analyzed by the Columbus software. Due to clustering of nuclear foci, which led to difficulties discerning between individual foci within a nucleus, the aggregate area subtended by all foci per nuclear area was measured instead. Each assay plate was internally normalized to the respective DMSO control data and the relative fold change to DMSO was averaged across replicate plates (Appendix A). The impact on foci area for all compounds screened along with change in nuclei count, serving as a proxy for cytotoxicity, can be found in supplementary Appendix A. In this study, DMSO was used at a concentration of 0.05%, and no effects of DMSO were observed. Any compound which reduced foci content by 30% or greater was considered a “hit” and subjected to secondary validation. A 30% threshold was set based on the effects of the DMPK ASO in these differentiated myoblasts. To avoid toxic compounds, any treatment that resulted in 50% or greater reduction in nuclei were omitted from the hit list. Using these metrics, bortezomib, a proteasome inhibitor; vorinostat, an HDAC-inhibitor; and gemcitabine, a deoxycytidine nucleoside, were identified as potential foci-reducers (Figure 1).

### 4.4. Forward Transfection of ASO

Differentiated myoblasts were transfected with ASO using Lipofectamine RNAi MAX (Invitrogen, Burlington, ON, Canada), employing a similar protocol to siRNA transfections [58]. Upon receipt, the control (Isis No. 549148) and DMPK (Isis No. 486178) ASOs [59] (Ionis Pharmaceutical) were diluted in nuclear-free H_2_O from lyophilized pellets to make 1 mM stocks; each stock was further aliquoted and stored at −80 °C. Control ASO (Isis No. 549148, Ionis Pharmaceutical) was used as a negative control. The ASOs were diluted at around 1 µM, relative to the final transfection volume (diluted ASO/lipofectamine, and differentiation media); and the lipofectamine was used at a ratio of 2 µL per 1 mL final transfection volume. The ASO and lipofectamine were first separately diluted in optiMEM reduced serum medium (Gibco, Burlington, ON, Canada) and incubated at RT for 5 min—ASO/optiMEM and lipofectamine/optiMEM dilutions each represented one-eighth of the final transfection volume. The diluted ASO and lipofectamine were then combined and incubated at RT for approximately 20 min to allow the ASO-lipid complex to form—this complex represented one-fourth of the final transfection volume. The nucleotide-lipid complex was then mixed with differentiation media and added to differentiated myoblasts. The treatments were performed at 37 °C in the presence of 5% CO_2_ for 24 h.

### 4.5. High-Throughput Screening and Analyzing (CUG)_exp_ Foci in DM1 Patient Cells

#### 4.5.1. Cell Growth, Treatment, and Staining

Proliferative myoblasts were plated in 384-well collagen I-coated CellCarrier-384 Ultra microplates (Perkin Elmer, Woodbridge, ON, Canada) and grown at 37 °C with 5% CO_2_; all wells, including peripheral columns and rows, were used. Collagen I-coating works similarly to ECM gel for improving cell adhesion and was used in place of ECM gel-coated 384-well plates as the latter are not commercially available. For low-throughput follow-up validation assessing CUG foci, cells were grown on ECM gel-coated 384-well falcon plates. Once 70–90% confluence was attained, cells were differentiated for six days. Stock compounds, constituted in DMSO (Sigma, Oakville, ON, Canada), were diluted in differentiation media and added to differentiated cells. DMSO vehicle was used as a negative control. DMPK ASO (Isis No. 486178, IONIS Pharmaceuticals) were forward transfected as positive controls for foci reduction [26]. One well corresponded to one treatment condition with technical replicates repeated across triplicate plates (*n* = 3).

Post-treatment, the cells were fixed in 4% formaldehyde at RT for 10–20 min, washed to remove the fixative, permeabilized in 70% ethanol (EtOH) at 4 °C, and incubated 3 × 5–10 min with 1× Phosphate Buffered Saline (PBS, Fisher Scientific, Waltham, MA, USA) at RT to rehydrate the cells. The samples were pre-hybridized with 40% formamide (Sigma, Oakville, ON, Canada) and 2× SSC (Sigma, Oakville, ON, Canada) for 10 min at RT. The cells were then hybridized with 0.3 ng/µL of the Alexa555-(CAG)_10_ probe (custom oligos from Invitrogen) diluted in hybridization buffer at 37 °C overnight for the purpose of fluorescence in situ hybridization (FISH) analysis. The hybridization buffer constituted of 40% formamide (Sigma, Oakville, ON, Canada), 2× SSC, 0.2% bovine serum albumin (BSA; Sigma, Oakville, ON, Canada), 1 mg/mL tRNA (Roche), 1 mg/mL single-stranded DNA from salmon testes (Sigma, Oakville, ON, Canada), 2 mM vanadyl ribonucleoside complex (VRC; Sigma, Oakville, ON, Canada), and 10% dextran sulfate (Sigma, Oakville, ON, Canada). Following hybridization, the wells were first washed 3 × 20 min in 40% formamide/2× SSC at 37 °C, then washed 3 × 5–10 min with 1× PBS (Fisher Scientific, Waltham, MA, USA) to remove the formamide and SSC prior to Hoechst staining. The nuclei were stained with 5 μg/mL of Hoechst 33,342 DNA stain (Sigma, Oakville, ON, Canada) diluted in 1× PBS (Fisher Scientific, Waltham, MA, USA) at RT for 10–20 min followed by 3 × 5 min washes in 1× PBS (Fisher Scientific, Waltham, MA, USA) at RT. The stained samples were stored in 50–75 μL of 1× PBS (Fisher Scientific, Waltham, MA, USA) at 4 °C before and after imaging.

The Alexa555-(CAG)_10_ probe (custom oligos from Invitrogen) was resuspended from a lyophilized pellet in nuclease-free Tris-EDTA (TE) buffer, pH 8.0 (Ambion, Waltham, MA, USA). The resuspended probe was further aliquoted into light-protective, dark tubes and stored long-term at −80 °C. Hoechst 33,342 (Sigma, Oakville, ON, Canada) was aliquoted into light-protective dark tubes and stored at 4 °C.

#### 4.5.2. Image Acquisition and Analysis

The plates were scanned using the Opera high-content screening system (Perkin Elmer, Woodbridge, ON, Canada) and 16–20 fields were captured per well to acquire a representative sample of images. Columbus software (Perkin Elmer, Woodbridge, ON, Canada) was used to analyze the foci content. Treatment data points were normalized to negative control samples within each assay plate; the normalized technical replicates from triplicate plates were then averaged. Cells were scored based on the total foci signal (number of pixels) to nuclear signal (number of pixels). A minimum signal intensity threshold was included as the lower end of detection to ensure that background signal was not included in quantification. For each plate, the foci signal and nuclear signal in each well was normalized to that of the DMSO negative control and represented as a fold-change; normalized foci area per nuclear and nuclei for each treatment, relative to DMSO negative control, were averaged across the respective triplicate plates.

### 4.6. In Vivo Validation of Vorinostat Treatment in has^LR^ Mice

Wild-type FVB/N, contrhasHSA^SR^ (short CTG repeat), and DM1 HSA^LR^ mice [60] were used in collaboration with Dr. Bernard Jasmin’s lab (University of Ottawa). DM1 HSA^LR^ mice were treated daily with 25 mg/kg or 50 mg/kg vorinostat (Cayman Chemical, Ann Arbor, MI, USA) or vehicle by i.p. injections. For in vivo studies, vorinostat stock was reconstituted at a maximum concentration of 7.5 mg/mL in 5% DMSO (Sigma, Oakville, ON, Canada), 300 mg/mL HPBCD (Cedarlane, Burlington, ON, Canada), and 0.9% NaCl (Fisher Scientific, Waltham, MA, USA); vehicle stock solutions comprised 5% DMSO (Sigma, Oakville, ON, Canada), 300 mg/mL HPBCD (Cedarlane, Burlington, ON, Canada), and 0.9% NaCl (Fisher Scientific). Mice were weighed weekly. 24 h after the final injection, mice were euthanized by 120 mg/kg euthanyl and cervical dislocation. Skeletal muscles were dissected (i) for cryostat sectioning by embedding in Tissue-Tek OCT compound (VWR, Mississauga, ON, Canada) and flash-freezing in isopentane cooled with liquid nitrogen and (ii) for molecular analysis by flash-freezing directly in liquid nitrogen for cryostat sectioning. Tissue staining and image analysis, as well as RNA extraction and PCR analysis, are described below. All treatments, sample processing and data analysis were conducted blindly. The control mice group was comprised of both wild-type FVB/N and HSA^SR^ mice.

### 4.7. Tissue Sectioning, Staining and Image Analysis of Mouse Skeletal Muscle

Mouse tissues from the vorinostat in vivo trials were processed for microscopy as previously described [35]. Tissues were sectioned using a Microm HM 500 M cryostat into 10 µm thick sections and collected on microscope glass slides (Fisher Scientific, Waltham, MA, USA). For central nucleation analysis, tissue slides were stained by hematoxylin and eosin staining, mounted using Permount mounting media (Fisher Scientific, Waltham, MA, USA), and covered with microscope cover glass no. 1.5 (Fisher Scientific, Waltham, MA, USA). Stained tissues were visualized at 20× magnification using the EVOS FL Auto 2 microscope (Thermo Scientific, Waltham, MA, USA) and manually counted using an Image J counter. For immunofluorescence staining of CLCN1 and laminin, tissue slides were fixed in 4% PFA/1× PBS for 10 min at RT, washed in 1× PBS, permeabilized in 2% Acetone in 1× PBS (chilled) for 5 min at RT, washed 3 × 5 min in 1× PBS at RT, blocked in blocking buffer (1× PBS, 5% goat serum, 0.3% Triton X-100) containing 0.3 M glycine for 1 h at RT, incubated with primary antibodies diluted in blocking buffer for 2 h at RT (anti-CLCN1 antibody (abcam, ab189857, Cambridge, UK) diluted at 1/50 and anti-Laminin antibody (Millipore, MAB 1905-1) diluted at 1/100), washed again for 3 × 5 min in 1× PBS at RT, incubated with secondary antibodies diluted in blocking buffer for 1 h at RT (Alexa594 anti-rabbit antibody (Invitrogen, A11012, Burlington, ON, Canada) and anti-rat antibody (Invitrogen, A21208, Burlington, ON, Canada), both diluted at 1/400), washed for 3 × 5 min in 1× PBS at RT, counterstained with 5 μg/mL of Hoechst 33,342 DNA stain (Sigma, Oakville, ON, Canada) diluted in 1× PBS and incubated at RT for 20min, washed 3 × 5 min in 1× PBS at RT, and finally, mounted with 1–2 drops of Dako fluorescence mounting medium (Agilent, S3023, Santa Clara, CA, USA) using no. 1.5 coverslips.

### 4.8. Protein Extraction, Quantification and Western Blotting

Myoblasts were seeded at 100,000–200,000 cells per well in 6-well plates (final volume of 2 mL); once cells reached a confluence of 70–90%, they were serum-starved to induce differentiation. Media was aspirated, the cells were washed with 1× PBS (Fisher Scientific, Waltham, MA, USA) to remove residual serum, the PBS was aspirated, and cells were harvested by either trypsinization or by scraping. To harvest by trypsinization, cells were lifted from the plate by incubating at 37 °C with 1× (0.25%) trypsin (Gibco); reconstituted in 15 mM NaCl (Fisher Scientific, Waltham, MA, USA), 0.5 mM EDTA (Sigma, Oakville, ON, Canada), and 1× PBS (Fisher Scientific, Waltham, MA, USA)) for 3–7 min. The trypsin was then inhibited using complete growth media, and the cell suspension was transferred to tubes and centrifuged at 300× *g* for 5–10 min. The supernatant was then removed, the cell pellet washed gently in 1× PBS (Fisher Scientific, Waltham, MA, USA) to remove residual serum, centrifuged further at 300× *g* for 5–10 min, and the final wash supernatant then removed, leaving the cell pellet. The cell pellet was then resuspended in RIPA (Radioimmunoprecipitation assay) lysis buffer containing a cocktail of protease and phosphatase inhibitors, diluted as per manufacturer’s instructions (Halt protease and phosphatase inhibitor cocktail, Thermo Scientific, Waltham, MA, USA), and incubated for 20–30 min at 4 °C. To lyse cells directly on-plate, the washed plate/cells were incubated on-plate for 20–30 min at 4 °C with lysis buffer; the lysed samples were then collected by scraping and transferred to tubes. The lysed cell suspensions were further subjected to sonication (8 × 15 s on, 60 s off) in a Bioruptor water bath sonicator (Diagenode, Denville, NJ, USA), and centrifuged at 13,000× *g* for 30 min at 4 °C; the supernatant containing the protein was recovered and retained. Protein concentrations were determined by Bradford protein assay using a Bio-Rad protein assay kit (Bio-Rad, Hercules, CA, USA) or a DC protein assay kit (Bio-Rad, Hercules, CA, USA), and BSA (Sigma, Oakville, ON, Canada) standards ranging from 0.125–2 μg/μL.

Loading samples were prepared in Laemmli buffer (Bio-Rad, Hercules, CA, USA) containing 1% β-mercaptoethanol (β-Me, Sigma, Hercules, CA, USA) and heated for 5 min at 95 °C. The samples were separated by a 10% SDS-PAGE gel prepared using the TGX stain-free FastCast acrylamide kit (Bio-Rad, Hercules, CA, USA), run at 200 V. The resolved protein gels were activated for 5 min under UV in the ChemiDoc imaging system (Bio-Rad, Hercules, CA, USA) in order to cross-link the stain-free molecule to the protein. The protein was then transferred to a low-fluorescence (LF) PVDF membrane using the Trans-Blot Turbo RTA LF PVDF transfer kit and the Trans-Blot Turbo Transfer System, as per manufacturer’s instructions (Bio-Rad, Hercules, CA, USA). The membranes were imaged for stain-free signal in the ChemiDoc (Bio-Rad) and blocked in blocking solution (5% milk, 1× TBS, 0.1% Tween-20) at RT for 30–60 min under agitation on a table-top rocker at low speed. MBNL1 mouse IgG primary antibody (Abcam) was diluted to 1:5000 in 1% milk in 1× PBS or TBS containing 10% sodium azide (NaAz) and incubated overnight at 4 °C, on a table-top rocker at low speed. HRP-linked anti-mouse secondary antibody (Bio-Rad, Hercules, CA, USA) was diluted 1:5000 in 5% milk and incubated for 1 h at RT. HRP-tagged antibody complexes were activated by incubation with enhanced luminol-based chemiluminescent (ECL) substrate. The chemiluminescence was visualized using a camera-based system (ChemiDoc Imaging System, BioRad, Hercules, CA, USA). Quantification was performed by densitometric analysis using the ImageLab software (BioRad, Hercules, CA, USA).

### 4.9. RNA Extraction and Polymerase Chain Reaction (PCR)

#### 4.9.1. RNA Extraction from Cell Culture

After treatment, media was aspirated, the cells were washed with 1× PBS (Fisher Scientific, Waltham, MA, USA) to remove residual serum and harvested as a cell pellet or directly on plate. To harvest as a cell pellet by trypsinization, cells were lifted from the plate by incubating at 37 °C with 1× (0.25%) trypsin (Gibco, Burlington, ON, Canada); reconstituted in 15 mM NaCl (Fisher Scientific, Waltham, MA, USA), 0.5mM EDTA (Sigma, Oakville, ON, Canada), and 1× PBS (Fisher Scientific, Waltham, MA, USA) for 3–7 min. The trypsin was then inhibited using media containing 10–30% FBS, and the cell suspension were transferred to tubes and centrifuged at 300× *g* for 5–10 min. The supernatant was then removed, the cell pellet was washed gently in 1× PBS (Fisher Scientific, Waltham, MA, USA) to remove residual serum, centrifuged further at 300× *g* for 5min, and the supernatant was removed, leaving the cell pellet. The cell pellet was then resuspended in lysis buffer and incubated for 5–10 min at RT to extract the RNA. To harvest on-plate, lysis buffer was added directly to the cells on-plate following the PBS wash and incubated at RT for 5–10 min. RNA extractions were done using the RNeasy micro kit with on-column Dnase treatment, as per the manufacturer’s protocol (Qiagen, Toronto, ON, Canada) with the following modifications: at each wash step with RW1 buffer, RPE buffer and 80% EtOH, columns were inverted 5–10 times to resuspend all contaminants; and for the last step of RNA elution, the columns were incubated with 20–22 μL of Rnase-free H_2_O for at least 5 min at RT. All RNA was quantified using a Nanodrop1000 spectrophotometer (Thermo Scientific, Waltham, MA, USA). For RNA experiments, myoblasts were seeded at 100,000–200,000 cells per well in 6-well plates (final volume of 2 mL); once cells reached a confluence of 70–90%, they were serum-starved to induce differentiation.

#### 4.9.2. RNA Extraction from Mouse Tissue

Upon dissection, the tissues were immediately flash-frozen in liquid nitrogen. The frozen tissues were crushed into a fine powder using a tissue pulveriser on dry ice, about 50–100 μg of the crushed tissue per sample was aliquoted, to which 1 mL of TRIzol (Thermo Fisher Scientific, Waltham, MA, USA) was added. The remaining extraction was done on-column using the Rneasy mini kit (Qiagen, Toronto, ON, Canada) or Purelink RNA mini kit/Trizol Plus RNA Purification Kit (Thermo Fisher Scientific, Waltham, MA, USA), with on-column Dnase treatment, as per the manufacturer’s instruction. All RNA was quantified using a Nanodrop1000 spectrophotometer (Thermo Scientific, Waltham, MA, USA).

#### 4.9.3. cDNA Synthesis

All cDNA was synthesized from RNA using reverse transcription (RT) at a 1:1 mRNA to cDNA synthesis ratio using the iScript Advanced cDNA synthesis kit (Bio-Rad, Hercules, CA, USA), as per manufacturer’s protocol. Each 20 μL RT reaction was thereafter diluted at least 10-fold in nuclease-free H_2_O before use in PCR reactions.

#### 4.9.4. Quantitative, Real-Time PCR (qPCR) for Steady-State mRNA Levels and Alternative Splicing

PCR reactions were set up as duplicates for each cDNA sample in hard-shell 96-well PCR plates (Bio-Rad, Hercules, CA, USA); each plate was sealed with Microseal ‘B’ PCR Plate Sealing Film (Bio-Rad, Hercules, CA, USA). For splicing assays, each 20 μL PCR reaction consisted of 1× iQ SYBR Green qPCR Supermix (Bio-Rad, Hercules, CA, USA), 0.5 μM FWD primer, 0.5 μM REV primer, and up to 50 ng of cDNA. For quantification of total transcribed mRNA for each gene, each 20 μL PCR reaction consisted of 1× iQ SYBR Green qPCR Supermix (Bio-Rad, Hercules, CA, USA), 0.25 μM FWD primer, 0.25 μM REV primer, and up to 50 ng of cDNA. The CFX96 Touch Real-Time PCR Detection System (Bio-Rad, Hercules, CA, USA) was used for qPCR reactions as well as SYBR green signal measurement; the general amplification protocol was 95.0 °C for 3 min, 40 × (95.0 °C for 10 s, 60.0 °C for 30 s, 72.0 °C for 30 s), 95.0 °C for 10 s. Following the amplification, a melt curve assay was also performed in order to assess purity and homogeneity of the amplified products: 65.0 °C to 95.0 °C, increment 0.5 °C for 5 s with readout of SYBR green signal. The raw data was further processed using the CFX Manager Software (Bio-Rad, Hercules, CA, USA) and converted to relative expression levels using the ΔΔCq formula [61]. All primers were optimized for use at 60 °C annealing temperature and the SybrGreen signal was acquired after extension at 72 °C. The SERCA1 splicing assay by qPCR has been previously described [58]. The primer sequences for human qPCR were (5′ to 3′) and they have shown in Table 1.

#### 4.9.5. Semi-Quantitative PCR (sqPCR) and qPCR for Alternative Splicing in Mouse Tissue

PCR reactions were done using 1× GoTaq Green Mastermix (Promega, WI, USA), 0.2 μM FWD primer, 0.2 μM REV primer, and up to 50 ng of cDNA (Table 2 and Table 3). The amplification protocol for was 95.0 °C for 2 min, 25 × (95.0 °C for 30 s, 55.0 °C for 30 s, 72.0 °C for 30 s), 72.0 °C for 2 min. The primer sequences are summarized below [35]. Following amplification, the samples were loaded on to a 5% non-denaturing polyacrylamide gel in 1× TBE for DNA gel electrophoresis and resolved at 200 V. The gel was then stained with 1/10,000 GelRed nucleic acid gel stain (VWR, Mississauga, ON, Canada) diluted in 1× TBE. The gels were imaged using the ChemiDoc imaging system (Bio-Rad) and quantified by densitometric analysis using ImageLab (Bio-Rad, Hercules, CA, USA).

### 4.10. Statistical Analyses

The mean and standard deviation are reported for continuous variables. Analysis of variance was performed using a two-way ANOVA test in order to determine the differences between the different treatment conditions. In this study, the alpha level of significance was set at 0.05. IBM SPSS Statistics for Windows, Version 25.0 (IBM Corp., Armonk, NY, USA) was used for statistical analysis.

## Figures and Tables

**Figure 1 ijms-24-03794-f001:**
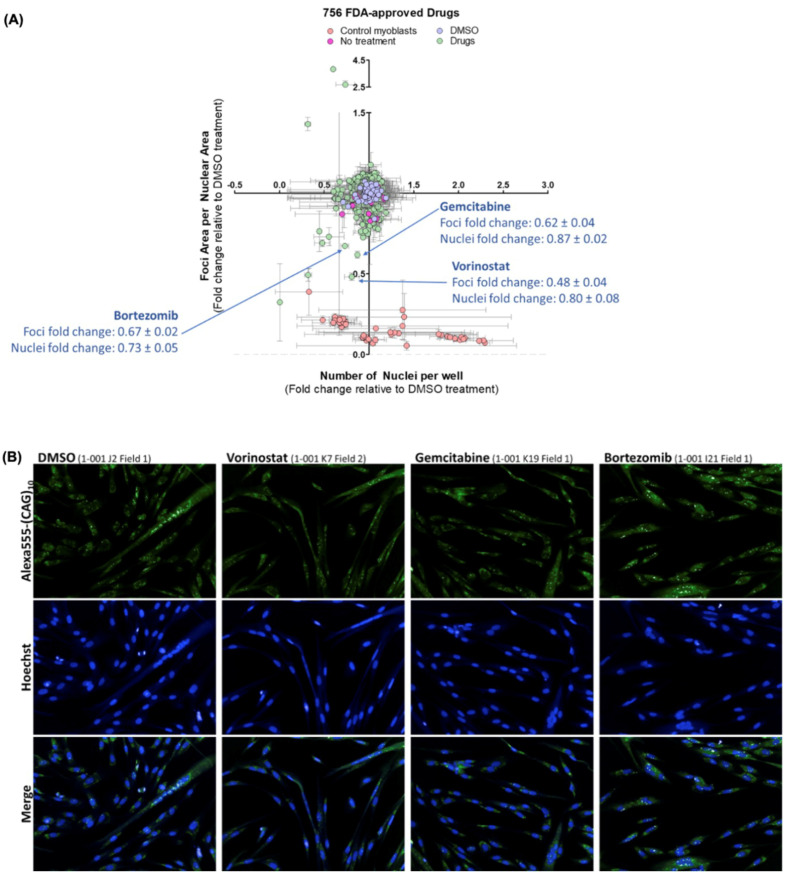
FDA small-molecule screen identified vorinostat, gemcitabine, and bortezomib as potential down-regulators of CUG foci. DM1_3600CTG_ myoblasts were serum-starved in 384-well plates for 7 days and treated with 2 μM of drugs for 24 h. Post-treatment, cells were stained with Hoechst for DNA and Alexa555-(CAG)_10_ for CUG RNA foci. Foci area per nuclear area and number of nuclei per well were quantified using Columbus and normalized to DMSO control data per plate. (**A**) Data is presented as the average fold-change relative to DMSO treatment (*n* = 3, error bars represent SD). (**B**) Representative images of top hits for small molecules which reduce foci. Full data set is summarized in Appendix A.

**Figure 2 ijms-24-03794-f002:**
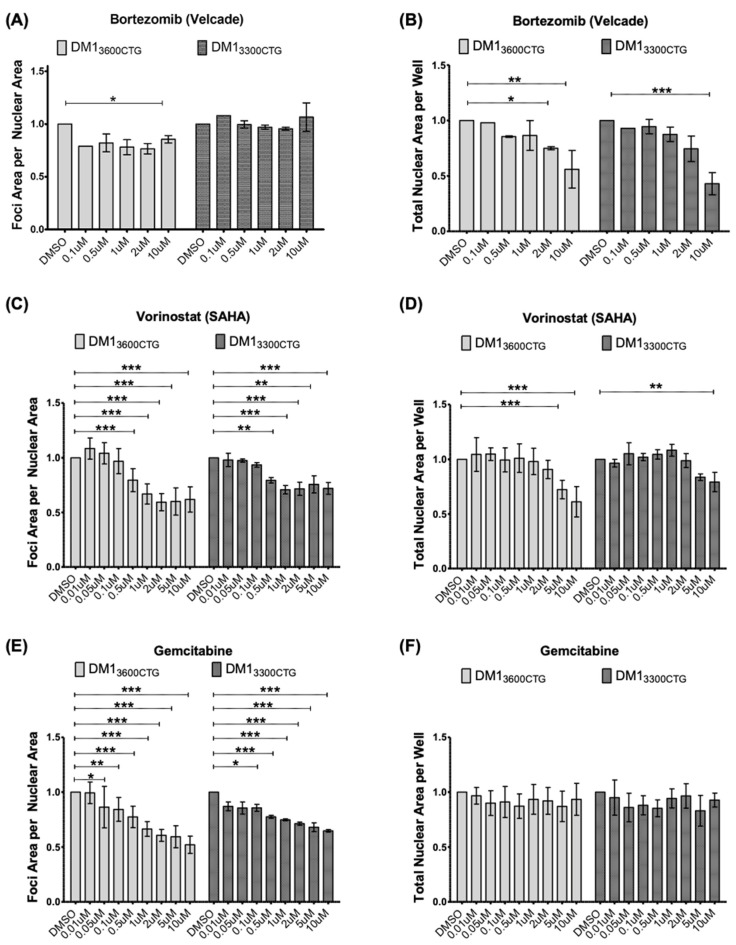
Bortezomib minimally reduced foci; vorinostat (SAHA) and gemcitabine reduced foci in both DM1_3600CTG_ and DM1_3300CTG_ differentiated myoblasts. DM1_3600CTG_ and DM1_3300CTG_ myoblasts were serum-starved in 384-well plates for 7 days and treated with 0.1–10 μM of (**A**,**B**) bortezomib, (**C**,**D**) vorinostat (SAHA), and (**E**,**F**) gemcitabine. Post-treatment, cells were fixed with 4% PFA, DNA was stained with Hoechst and CUG RNA foci were probed by Alexa555-(CAG)_10_ fluorescent oligo. (**A**,**C**,**E**) Foci area per nuclear area and (**B**,**D**,**F**) total nuclear area per well (to assess treatment-associated toxicity) were quantified using Columbus and normalized to DMSO control; data is presented as fold-change relative to DMSO treatment (*n* ranges from 1 (for Bortezomib 0.1 µM) to 8, two-way ANOVA; error bars represent SD). * *p* < 0.05, ** *p* < 0.01, *** *p* < 0.001.

**Figure 3 ijms-24-03794-f003:**
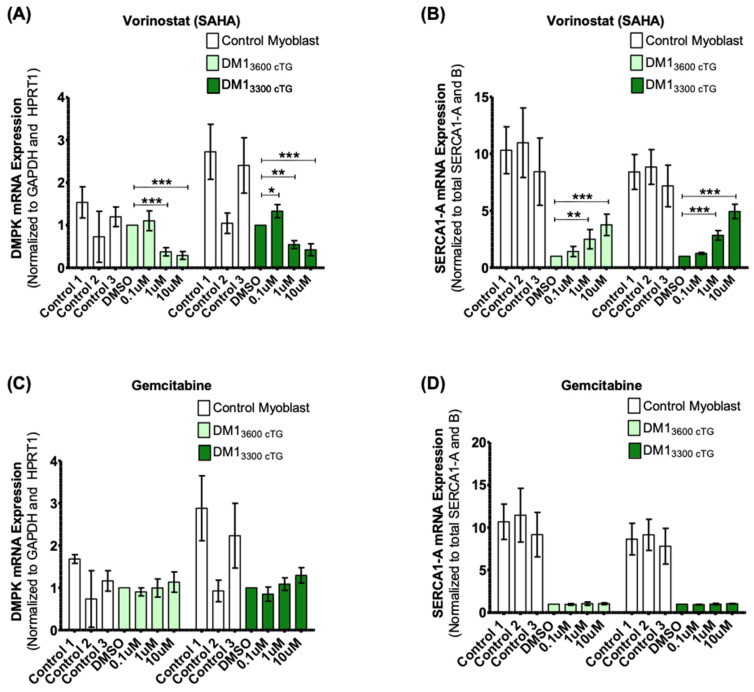
Vorinostat (SAHA), but not gemcitabine, reduced *DMPK* mRNA levels and rescued the *SERCA1-A* splice product in both DM1_3600CTG_ and DM1_3300CTG_ differentiated myoblasts. Control DM1_3600CTG_ and DM1_3300CTG_ myoblasts were serum-starved in 6-well plates for 7 days. DM1 cells were treated with DMSO alone or 0.1, 1 or 10 μM of (**A**,**B**) vorinostat (SAHA) or (**C**,**D**) gemcitabine; control myoblasts were treated with DMSO and used for baseline quantification in unaffected cells. RNA was extracted (RNeasy micro kit, Qiagen) and reverse-transcribed to cDNA (iScript Advanced RT kit, Biorad). RT-qPCR was performed (iQ Sybr green supermix, Biorad) to assess mRNA levels (*n* = 6, two-way ANOVA; error bars represent SD). * *p* < 0.05, ** *p* < 0.01, *** *p* < 0.001.

**Figure 4 ijms-24-03794-f004:**
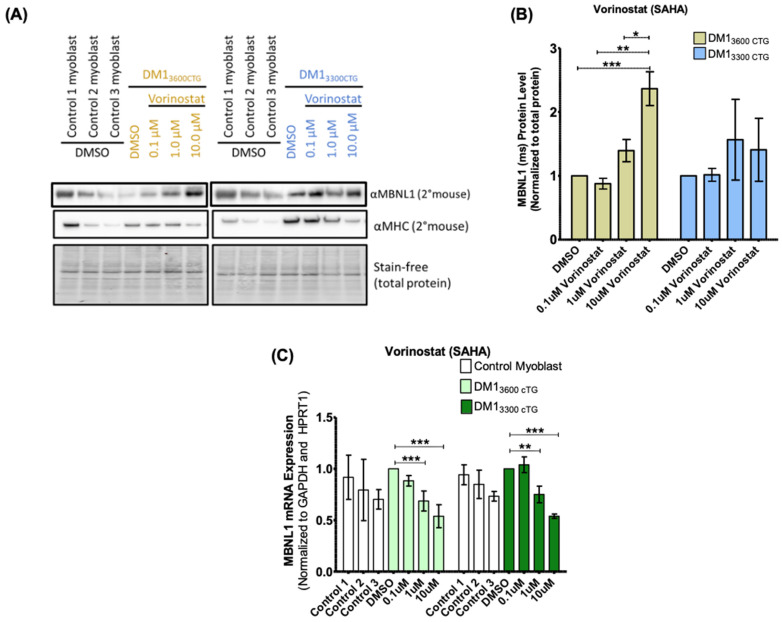
Vorinostat increased MBNL1 protein in DM1_3600CTG_ differentiated myoblasts but decreased *MBNL1* mRNA. (**A**,**B**) DM1_3600CTG_ and DM1_3300CTG_ myoblasts were serum-starved in 6 cm plates for 7 days and treated with DMSO alone or 0.1, 1 and 10 μM of vorinostat (SAHA). 24 h post-treatment, cells were trypsinized and extracted for protein using RIPA lysis buffer for western blot analysis. (**A**) Western blot images; (**B**) quantification of MBNL1 protein normalized to total protein and presented as fold-change relative to DMSO control (*n* = 2; two-way ANOVA). (**C**) DM1_3600CTG_ and DM1_3300CTG_ myoblasts were serum-starved in 6-well plates or 6 cm plates and treated with DMSO alone or 0.1, 1 and 10 μM of vorinostat (SAHA). 24 h post-treatment, cells were lysed directly on plate (6-well plate) or trypsinized and the pellet lysed (6 cm plate). RNA was extracted (RNeasy micro kit, Qiagen) and reverse-transcribed to cDNA (iScript Advanced RT kit, Biorad). RT-qPCR was performed (iQ Sybr green supermix, Biorad) to assess mRNA levels (*n* = 6, two-way ANOVA; error bars represent SD). * *p* < 0.05, ** *p* < 0.01, *** *p* < 0.001.

**Figure 5 ijms-24-03794-f005:**
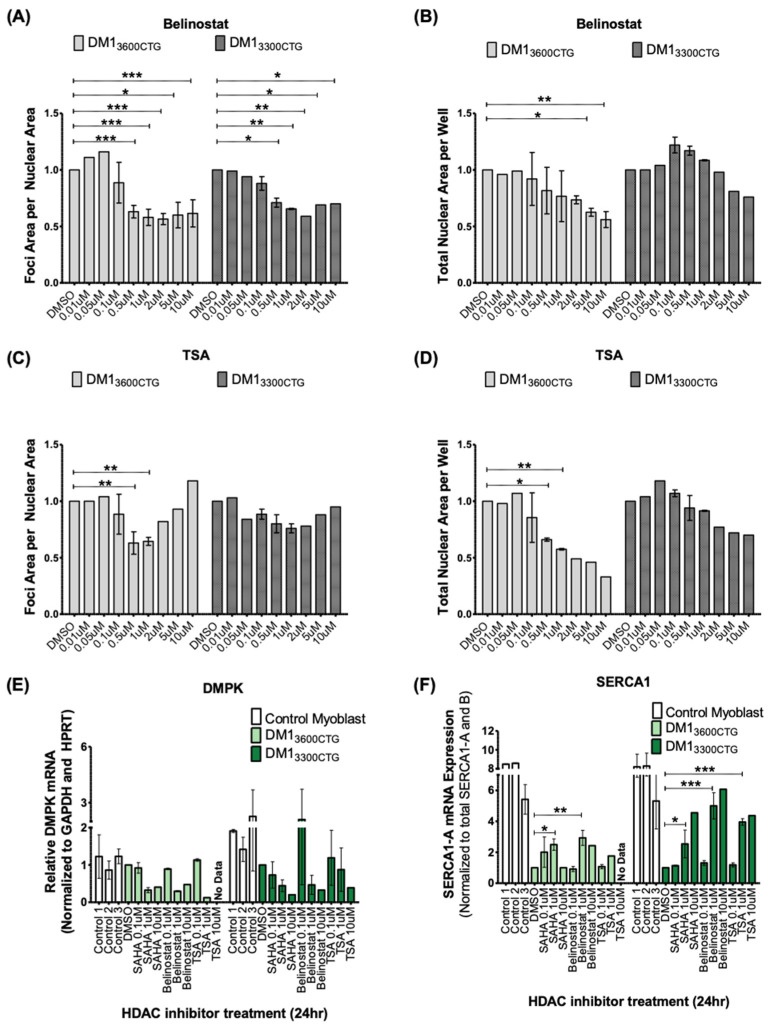
Two additional pan-HDAC inhibitors, belinostat and trichostatin A (TSA), were validated to reduce foci, decrease *DMPK* mRNA, and rescue *SERCA1* splicing in differentiated DM1 myoblasts. (**A**–**D**) DM1_3600CTG_ and DM1_3300CTG_ myoblasts were grown and/or serum-starved in 384-well plates for 7 days and treated with DMSO alone or 0.01, 0.05, 0.1, 0.5, 1, 2, 5 and 10 μM of HDAC inhibitor. 24 h post-treatment, cells were fixed with 4% PFA, DNA was stained with Hoechst and CUG RNA foci were probed by Alexa555-(CAG)_10_ fluorescent oligo. Foci area per nuclear area and total nuclear area per well (to assess treatment-associated toxicity) were quantified using Columbus and normalized to DMSO control; data is presented as fold-change relative to DMSO treatment (*n* ranges from 1 to 3, two-way ANOVA; error bars represent SD). (E-F) DM1_3600CTG_ and DM1_3300CTG_ myoblasts were serum-starved in 6-well plates for 7 days. DM1 cells were treated with DMSO alone or 0.1, 1 or 10 μM of vorinostat (SAHA), belinostat, and TSA. RNA was extracted (RNeasy micro kit, Qiagen) and reverse-transcribed to cDNA (iScript Advanced RT kit, Biorad). RT-qPCR was performed (iQ Sybr green supermix, Biorad) to assess mRNA levels of (**E**) DMPK and (**F**) SERCA1 splicing. (*n* ranges from 1 to 2, two-way ANOVA; error bars represent SD). * *p* < 0.05, ** *p* < 0.01, *** *p* < 0.001.

**Figure 6 ijms-24-03794-f006:**
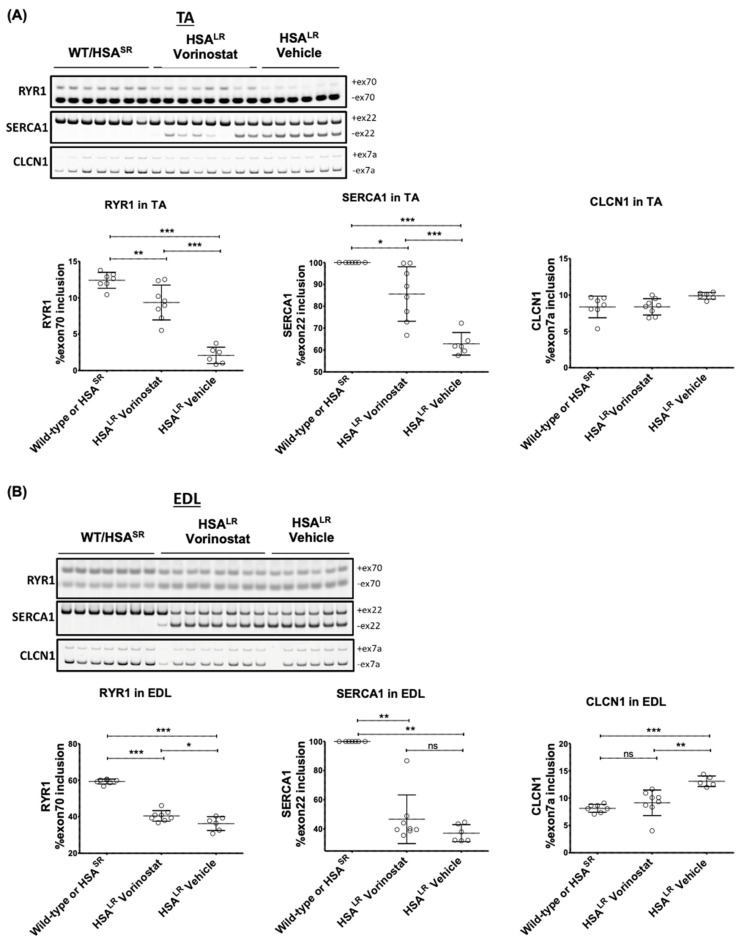
Vorinostat (SAHA) showed promising therapeutic effects in the DM1 HSA^LR^ mouse model, with greater impact in TA than EDL. Mice at approximately 4 weeks of age were injected (IP) daily with vehicle or 50 mg/kg vorinostat (SAHA) for 4 weeks. The mice were sacrificed by lethal injection and cervical dislocation. Skeletal muscle tissue from one hind leg was frozen in OCT for sectioning and imaging, and skeletal muscle tissue from the other hind leg was flash frozen in liquid nitrogen for RNA workup. Analysis of DM1 related spliceopathy. Flash frozen tissue was ground to a powder and a portion used for Trizol RNA extraction using 5 the Purelink RNA mini kit (Invitrogen) and reverse transcribed to cDNA (iScript Advanced RT kit, BioRad). RT-sqPCR products using cDNA from (**A**) TA and (**B**) EDL were resolved on 7% acrylamide gel, stained with Gel Red and imaged using the ChemiDoc (BioRad). Quantification of transcript ratios was done using ImageLab (BioRad). (*n* ranges from 6 to 8, two-way ANOVA; error bars represent SD). * *p* < 0.05, ** *p* < 0.01, *** *p* < 0.001. Abbreviations, EDL, *extensor digitorum longus*; TA, *tibialis anterior*.

**Figure 7 ijms-24-03794-f007:**
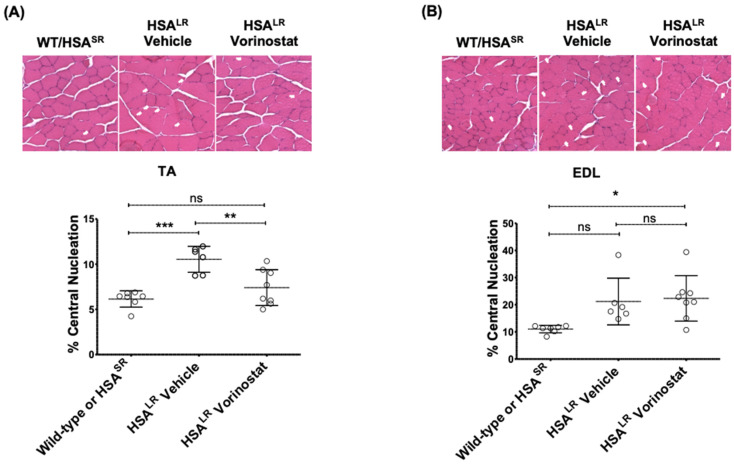
Vorinostat (SAHA) reduced central nucleation in TA muscle of DM1 HSA^LR^ mouse model. Mice at approximately 4 weeks of age were injected (IP) daily with vehicle or 50 mg/kg vorinostat (SAHA) for 4 weeks. The mice were sacrificed by lethal injection and cervical dislocation. Skeletal muscle tissue from one hind leg was frozen in OCT for sectioning and imaging, and central nucleation was assessed in (**A**) TA and (**B**) EDL muscle. Tissues flash frozen in OCT were sectioned at 10 μm thickness and subjected to H&E staining. Brightfield images were taken at 20× using EVOS cell imaging system and whole sections were manually counted for central nucleation. (*n* ranges from 6 to 8, two-way ANOVA; error bars represent SD). * *p* < 0.05, ** *p* < 0.01, *** *p* < 0.001. Abbreviations, EDL, *extensor digitorum longus*; ns, not significant; TA, *tibialis anterior*.

**Figure 8 ijms-24-03794-f008:**
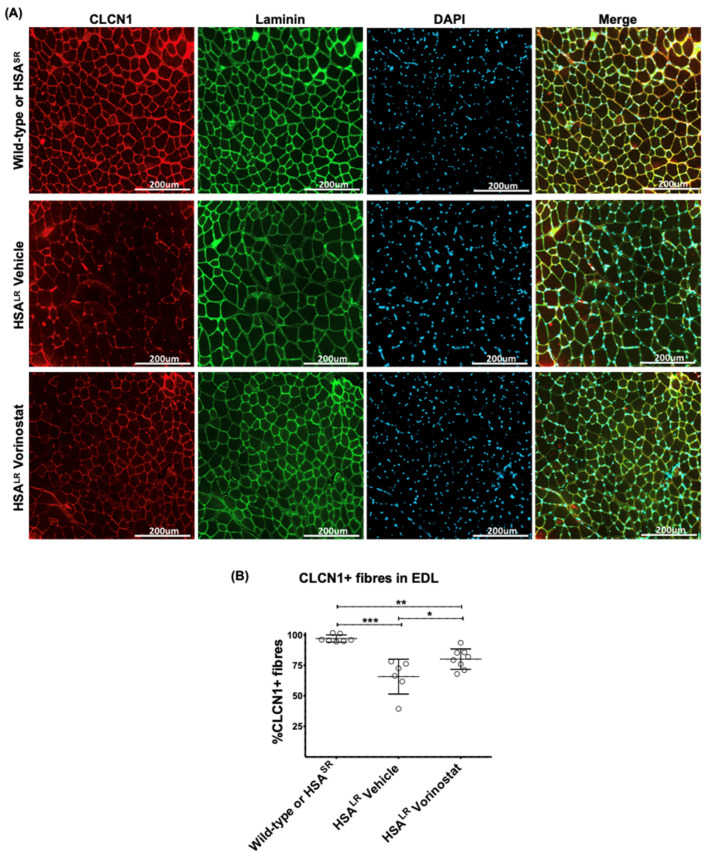
Vorinostat (SAHA) restored chloride channel protein levels in EDL muscle of DM1 HSA^LR^ mouse model. Mice at approximately 4 weeks of age were injected (IP) daily with vehicle or 50 mg/kg vorinostat (SAHA) for 4 weeks. The mice were sacrificed by lethal injection and cervical dislocation. Skeletal muscle tissue from one hind leg was frozen in OCT for sectioning and imaging. (**A**) Tissues flash frozen in OCT were sectioned at 10 μm thickness and subjected to immunofluorescence staining using antibodies against chloride channels (CLCN1) and laminin, and counterstained with DAPI. Brightfield images were taken at 20× using EVOS 5 cell imaging system. (**B**) CLCN1-positive fibers and lamini-positive fibers were manually counted using ImageJ and the percentage of CLCN1-positive fibers was calculated relative to total fibers as represented by lamini-positive fibers. (*n* ranges from 6 to 8, two-way ANOVA; error bars represent SD). * *p* < 0.05, ** *p* < 0.01, *** *p* < 0.001. Abbreviations, EDL, *extensor digitorum longus*.

**Table 1 ijms-24-03794-t001:** Primer sequences for human qPCR.

DMPK	Forward	GGCTCACTGCCATGGTGA
	Reverse	GCTGTTTCATCCTGTGGGGA
MBNL1	Forward	TGATTGTCGGTTTGCTCATC
	Reverse	TTGATCTTGGCTTGCAAATG
GAPDH	Forward	TGCACCACCAACTGCTTAGC
	Reverse	GCATGGACTGTGGTCATGAG
HPRT	Forward	TGACACTGGCAAAACAATGCA
	Reverse	GTCCTTTTCACCAGCAAGCT
SERCA1-A	Forward	CCCTCCTCCATCTCTGAGC
	Reverse	GCTCTGCCTGAAGATGTGTC
SERCA1-AB	Forward	CTCCATCTGCCTCTCCATGT
	Reverse	CTTGAGGACCATGAGCCACT

**Table 2 ijms-24-03794-t002:** Primer sequences for mouse sqPCR.

SERCA1	Forward	ATCTTCAAGCTCCGGGCCCT
	Reverse	CAGCTTTGGCTGAAGATGCA
RYR1	Forward	GACAATAAGAGCAAAATGGC
	Reverse	CTTGGTGCGTTCCTGATCTG
CLCN1	Forward	GGAATACCTCACACTCAAGGCC
	Reverse	CACGGAACACAAAGGCACTGAATGT

**Table 3 ijms-24-03794-t003:** Primer sequences for mouse qPCR.

has	Forward	GTGGATCACCAAGCAGGAGT
	Reverse	GTCAGTTTACGATGGCAGCA
mmGAPDH	Forward	CGTCCCGTAGACAAAATGGT
	Reverse	CTCCTGGAAGATGGTGATGG
mmHPRT	Forward	GCAAACTTTGCTTTCCCTGGTT
	Reverse	CAAGGGCATATCCAACAACA

## Data Availability

Data is available from corresponding authors upon reasonable request.

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
