# Peer review of "Vorinostat Improves Myotonic Dystrophy Type 1 Splicing Abnormalities in DM1 Muscle Cell Lines and Skeletal Muscle from a DM1 Mouse Model"

_ijms, 2023, doi:10.3390/ijms24043794_

Round 1

Reviewer 1 Report

Neault et al. in their study screened the library of FDA-approved small compounds as potential DM1 therapeutics. The authors treated two DM1 myocyte cell lines with three selected molecules and tested for RNA foci reduction, DMPK and MBNL1 mRNA and protein levels, also splicing correction. The authors provided a mechanistic insight by mimicking the vorinostat therapeutic effect by two additional histone deacetylase inhibitors. Finally, the efficacy of the vorinostat was recapitulated in the HSA-LR mouse model. There are a few clinical trials for DM1 in progress now, and no approved cure for this debilitating disease. Thus, the identification of already FDA-approved drugs as potential DM1 therapeutics is critical. The authors addressed this urgent clinical need in their study. The manuscript clearly describes the study, and I have only one major concern and a few cosmetic changes are required.

Major comment:

1. Figure 4. The difference in endogenous MBNL1 protein expression levels between control and DM1 myotubes looks like an experimental artefact. One can speculate a CUGexp/MBNL1 extraction/precipitation issue. The increase of MBNL1 protein detection in treated DM1 cells could reflect the reduction of sequestration/improved extractability. This concern is supported by the lack of differences at the MBNL1 mRNA expression levels.

Minor comments:

2. The Result section contains experimental procedure descriptions. The storyline can be improved by removing unnecessary text.

3. Ref.2 indicates the prevalence of DMPK CTGexp mutation in newborns and not DM1.

4. CUGexp RNA forms a hairpin structure in vitro. We do not know what the DMPK CUGexp mRNA structure is in vivo.

5. MBNL1 stands for Muscleblind Like Splicing Regulator 1.

6. Several text corrections: italicize gene names and Latin terms, make space between numbers and units, use Greek symbols for units, use consistently super/subscripts in names, remove the red wavy line (Figure 8), etc.

Author Response

Dear Dr. Jidapha Techataratip

Assistant Editor,

Thank you for providing us the opportunity to submit a revised version of our manuscript entitled Vorinostat Shows Improves Myotonic Dystrophy Type 1 Splicing Abnormalities in DM1 Muscle Cell Lines and Skeletal Muscle From a DM1 Mouse Model” to the International Journal of Molecular Sciences.

We thank the three reviewers for their thoughtful comments and suggestions regarding our manuscript. We have taken into account all of the comments and incorporated them into the revised manuscript. An itemized point-by-point response to the comments from reviewers is provided below in response to the changes made to the original document.

COMMENTS FROM REVIEWER 1

Comment #1

Neault et al. in their study screened the library of FDA-approved small compounds as potential DM1 therapeutics. The authors treated two DM1 myocyte cell lines with three selected molecules and tested for RNA foci reduction, DMPK and MBNL1 mRNA and protein levels, also splicing correction. The authors provided a mechanistic insight by mimicking the vorinostat therapeutic effect by two additional histone deacetylase inhibitors. Finally, the efficacy of the vorinostat was recapitulated in the HSA-LR mouse model. There are a few clinical trials for DM1 in progress now, and no approved cure for this debilitating disease. Thus, the identification of already FDA-approved drugs as potential DM1 therapeutics is critical. The authors addressed this urgent clinical need in their study. The manuscript clearly describes the study, and I have only one major concern and a few cosmetic changes are required. Major comment: Figure 4. The difference in endogenous MBNL1 protein expression levels between control and DM1 myotubes looks like an experimental artefact. One can speculate a CUGexp/MBNL1 extraction/precipitation issue. The increase of MBNL1 protein detection in treated DM1 cells could reflect the reduction of sequestration/improved extractability. This concern is supported by the lack of differences at the MBNL1 mRNA expression levels.

Response: Thank you for this, if we understand correctly the reviewer is wondering if DM1 myoblasts have higher MBNL1 protein levels than the western blots suggest because of sequestration of the MBNL1 protein in the expanded CUGn structure even after we extract for protein.  In fact, we have had the same thought and explored the possibility when we were extracting the cell protein, we wondered if the mild RIPA buffer we used wasn’t touching the MBNL1 protein sequestered in DMPK mRNA foci.  As intuitively appealing as the idea of MBNL1 sequestration was, our exploration with more powerful Urea detergent  buffer shows that in fact this is not the case, we did not see the increase in MBNL1 we would have predicted. figure attached.  

We can add this to supplementary data if useful.

Urea = extracted straight in urea

F1 = cell pellet extracted in RIPA as we would normally do protein extracted and the supernatant/soluble protein collected

F2 = leftover pellet after F1 supernatant was collected; this pellet was re-extracted with more RIPA and supernatant collected

F3 = leftover pellet after F2 supernatant was collected; this pellet was re-extracted with more RIPA and supernatant collected

F4 = leftover pellet after F3 supernatant was collected; this pellet was re-extracted one last time, this time with UREA

Comment #2

The Result section contains experimental procedure descriptions. The storyline can be improved by removing unnecessary text.

Response: Thank you. We have reviewed the text and removed unnecessary text in the results.

Comment #3

Ref.2 indicates the prevalence of DMPK CTGexp mutation in newborns and not DM1.

Response: We have rewritten the text to acknowledge this in the introduction section.

Comment #4

CUGexp RNA forms a hairpin structure in vitro. We do not know what the DMPK CUGexp mRNA structure is in vivo.

Response: Thanks, the text has been corrected and now states in vitro.

Comment #5

MBNL1 stands for Muscleblind Like Splicing Regulator 1.

Response: Thanks to the reviewer, the information was added.

Comment #6

Several text corrections: italicize gene names and Latin terms, make space between numbers and units, use Greek symbols for units, use consistently super/subscripts in names, remove the red wavy line (Figure 8), etc.

Response: Thanks to the reviewer, all the mistakes were corrected according to his/her comments.

Reviewer 2 Report

Well performed study to unravel the role of some HDAC inhibitors in myoblasts derived from DM1 patients. However some concerns:

What are the targets of vorinostat? are these targets affected in DM1 patients/myoblasts? are rescued after vorinostat treatment?

Why a decrease in DMPK mRNA levels is positive after vorinostat treatment?

HSAlr mice treated with vorinostat has longer survival?

Some minor points:

Supl 1B and 1C: it is possible to show a picture of the RNA fish in control myoblasts?

Table S1: is it important to show the well name and the plate id?

Fig 4A: it seems that MBNL is overexposed in control myoblasts. Is it possible to remove the red mark that shows that it is overexposed?

Author Response

Dear Dr. Jidapha Techataratip

Assistant Editor,

Thank you for providing us the opportunity to submit a revised version of our manuscript entitled Vorinostat Shows Improves Myotonic Dystrophy Type 1 Splicing Abnormalities in DM1 Muscle Cell Lines and Skeletal Muscle From a DM1 Mouse Model” to the International Journal of Molecular Sciences.

We thank the three reviewers for their thoughtful comments and suggestions regarding our manuscript. We have taken into account all of the comments and incorporated them into the revised manuscript. An itemized point-by-point response to the comments from reviewers is provided below in response to the changes made to the original document.

COMMENTS FROM REVIEWER 2

Comment #1

Well performed study to unravel the role of some HDAC inhibitors in myoblasts derived from DM1 patients. However some concerns: What are the targets of vorinostat? are these targets affected in DM1 patients/myoblasts? are rescued after vorinostat treatment?

Response: The targets of vorinostat are many and varied with many hundreds of  genes impacted [1]. Thus a clear sense of what the target is unknown.  The picture is even more complicated as  an earlier article showing an impact of vorinostat on MBNL1 wherein the authors state “the identification of several HDAC inhibitors that increased MBNL1 suggests a regulatory role of HDACs in MBNL1 expression. However, we recognize that the observed activity at the effective concentration ∼1 μM was probably due to inhibition of multiple HDAC family members rather than one sub-type” [2].

References

  1. Li, W.; Sun, Z. Mechanism of Action for HDAC Inhibitors—Insights from Omics Approaches. Int. J. Mol. Sci. 2019, 20, 1616, doi.10.3390/ijms20071616.
  2. Zhang F, Bodycombe NE, Haskell KM, Sun YL, Wang ET, Morris CA, Jones LH, Wood LD, Pletcher MT. A flow cytometry-based screen identifies MBNL1 modulators that rescue splicing defects in myotonic dystrophy type I. Human molecular genetics. 2017 Aug 15;26(16):3056-68.

Comment #2

Why a decrease in DMPK mRNA levels is positive after vorinostat treatment?

Response: If we understand the reviewers comments correctly, she/he is asking why DMPK mRNA levels are decreased by vorinostat.  Simply put, as we state in the introduction “The greater the number of CTG repeats, the more severe the symptoms and the earlier the age of onset [14]. The most widely accepted DM1 pathogenic model invokes RNA gain-of-function with pathogenically expanded DMPK mRNA forming hairpin structures through C-G base-pairing [15,16] which aggregate into nuclear foci and cause misregulation and/or sequestration of a number of RNA-binding proteins (RBP) [17].” Thus, a priori, we believe that treatments that decrease DMPK likely would have a favorable effect on DM1 as we have observed here for vorinostat.

Comment #3

HSALR mice treated with vorinostat has longer survival?

Response: We did not formally assess the impact of vorinostat on survival. For the four-week period that the approximately I month old mice were assessed while dosing with drug, two of the eight control treated mice died while all eight of the SAHA treated mice survived (Figure S6A). This is too small a number and too short a time to say anything about the impact of vorinostat on mouse survival.

Comment #4

Supl 1B and 1C: it is possible to show a picture of the RNA fish in control myoblasts?

Response: Thanks to the reviewer for his/her comment, the Figure S1 was modified adding showing the foci in the control myoblasts.

Comment #5

Table S1: is it important to show the well name and the plate id?

Response: Thank you for your comment. We have removed these, S1 Table edited  

Comment #6

Fig 4A: it seems that MBNL is overexposed in control myoblasts. Is it possible to remove the red mark that shows that it is overexposed?

Response: Figure 4 was modified as requested.

Reviewer 3 Report

In this paper the authors screened an FDA-approved drug library of small molecules using two immortalized myoblasts obtained from form patients with DM1 disease (Euro biobank). The readout was the reduction of CUG foci. The authors ultimately found few drugs that corrected DM1 spliceopathy, the hallmark of DM1 disease. The authors identified vorinostat an HDAC inhibitor as the most effective drug to reduce in vitro and in vivo DM1 phenotypes. The treatment was also effective in restoring the spliceopathy of SERCA1 and cell surface expression of a chloride channel in skeletal muscle cells. The author concluded that vorinostat could be used as a potential treatment for DM1 disease.

The paper is potentially of interest but there are several issues which limited the enthusiasm for the manuscript. The major concern is the threshold level set for inclusion or exclusion of the most effective drug and the adequacy of the mouse model used. Specifically:

- The authors decided to omit any treatment that resulted in 50% or greater reduction in nuclei from the hit list and instead used the 30% cut off based on ASO data. The concern is that several potentially beneficial FDA approved drugs may have been eliminated based on ASO data derived from a different mechanism of action compared to small drugs. For example, what happen to the drugs that were between 30% and 50%? Are there any data that can be used for comparison? This is a major concern.

-How was toxicity evaluated as this will impact the hit list selection?

-The mouse model for DM1 used is overexpressing only 250 CTG repeats compared with cells lines which have much higher CTG repeats. The authors should consider the mouse model of DM1 such as DMSXL which expresses more than 1000 CTG repeat and more comparable to the cell lines used.

- All compounds in the FDA-approved small-molecule screen were assayed at a final

concentration of 2µM in vitro and 25-50 mg/kg in vivo. How were these doses selected and what was the calculation from in vitro to in vivo studies?

- Indicate the number of animals and cells used in each experiment.

-In the introduction the authors referred to the beneficial effect of ASO as a treatment for the disease however the references shown are misleading, since they do not specifically address this issue. There are recent papers that were published showing this beneficial effect on skeletal muscle and the brain. Ait Benichou et al., 2021 and 2022.

-The authors should refer to the work in which ISIS486178 was first described, used here as a control.

-The authors should clearly state what was the concentration (or the volume%) of DMSO used in control treatments and what was the concentration of DMSO in 10microM concentration of the drugs, the highest used in the study. The data was normalized to DMSO treatments. The question is, did the DMSO has any effects? And at what concentration this was studied?

-Vorinostat reduced the DMPK mRNA levels (Figure 3), is this the mechanism of action of the drug?

-Figure 4A and Figure 4, in the supplementary data showing αMBNL1 expression the mouse where many lines are saturated and therefore unacceptable.

-The conclusion has to be revised since the disease is multisystemic. The authors did not show that the drug can be used to affect other organs such as the brain and the heart the most problematic aspect of the disease. Hence the conclusion and title need be to revised and reflect outcomes on skeletal muscle only.

Author Response

Dear Dr. Jidapha Techataratip

Assistant Editor,

Thank you for providing us the opportunity to submit a revised version of our manuscript entitled Vorinostat Shows Improves Myotonic Dystrophy Type 1 Splicing Abnormalities in DM1 Muscle Cell Lines and Skeletal Muscle From a DM1 Mouse Model” to the International Journal of Molecular Sciences.

We thank the three reviewers for their thoughtful comments and suggestions regarding our manuscript. We have taken into account all of the comments and incorporated them into the revised manuscript. An itemized point-by-point response to the comments from reviewers is provided below in response to the changes made to the original document.

COMMENTS FROM REVIEWER 3

Comment #1

In this paper the authors screened an FDA-approved drug library of small molecules using two immortalized myoblasts obtained from form patients with DM1 disease (Euro biobank). The readout was the reduction of CUG foci. The authors ultimately found few drugs that corrected DM1 spliceopathy, the hallmark of DM1 disease. The authors identified vorinostat an HDAC inhibitor as the most effective drug to reduce in vitro and in vivo DM1 phenotypes. The treatment was also effective in restoring the spliceopathy of SERCA1 and cell surface expression of a chloride channel in skeletal muscle cells. The author concluded that vorinostat could be used as a potential treatment for DM1 disease. The paper is potentially of interest but there are several issues which limited the enthusiasm for the manuscript.

Response: Thanks to the reviewer for his/her comment about our manuscript.

Comment #2

The major concern is the threshold level set for inclusion or exclusion of the most effective drug and the adequacy of the mouse model used. Specifically: The authors decided to omit any treatment that resulted in 50% or greater reduction in nuclei from the hit list and instead used the 30% cut off based on ASO data. The concern is that several potentially beneficial FDA approved drugs may have been eliminated based on ASO data derived from a different mechanism of action compared to small drugs. For example, what happen to the drugs that were between 30% and 50%? Are there any data that can be used for comparison? This is a major concern.

Response:  We appreciate and thank reviewer 1 for his/her comments. The main criterion used to establish the threshold of effectiveness in reducing nuclear foci, as explained in the manuscript, was to obtain results similar to those already obtained with DMPK ASO (ISIS486178, Ionis Pharmaceuticals). We believe that using a reduction of more than 50% might lead to identification of compounds with potential toxicity. Nonetheless, we do agree with the reviewer that ultimately evaluation of compounds that reduce foci by 30 to 50% as part of our future research would be of interest. For the time being, we believe the demonstration efficacy of vorinostat both in vitro and in vivo as a potential treatment for DM1 is a worthwhile observation. Moreover, we believe that our approach is validated by the fact that other HDAC inhibitors, such as belinostat and trichostatin, show similar efficacy in reducing nuclear foci.

Comment #3

-How was toxicity evaluated as this will impact the hit list selection?

Response: As stated in the text “To avoid toxic compounds, any treatment that resulted in 50% or greater reduction in nuclei were omitted from the hit list.’

Comment #4

The mouse model for DM1 used is overexpressing only 250 CTG repeats compared with cells lines which have much higher CTG repeats. The authors should consider the mouse model of DM1 such as DMSXL which expresses more than 1000 CTG repeat and more comparable to the cell lines used.

Response: Although the DM1 mouse model used is well accepted and has been widely used for pre-clinical testing of therapeutic candidates for ASO therapy [24] as well as small molecules (more recently in https://doi.org/10.1186/s12967-022-03806-9), we agree that ultimately testing SAHA on the DMSXL model would be of interest. 

Comment #5

- All compounds in the FDA-approved small-molecule screen were assayed at a final concentration of 2 µM in vitro and 25-50 mg/kg in vivo. How were these doses selected and what was the calculation from in vitro to in vivo studies?

Response: In accordance with the manufacturer's recommendations, all compounds were screened at a final concentration of 2 µM in the FDA-approved small-molecule screen. In vivo doses were determined based on vorinostat animal data, where doses up to 50 and 150 mg/kg/day were tested in rats and rabbits, respectively.

Comment #6

- Indicate the number of animals and cells used in each experiment.

Response: Using the reviewer comment, the number of animals were added and highlighted in yellow. The number of cells is as follows:

Figures 1, 2 and 5, for foci assay in 384-well plates, myoblasts were seeded at approximately 2000-3000 cells per well (final media volume ranged from 30-40 µl per well); once cells reached a confluence of 70-90%, they were serum-starved to induce differentiation.

Figures 3 and 4, for RNA and protein experiments, myoblasts were seeded at 100,000-200,000 cells per well in 6-well plates (final volume of 2 ml); once cells reached a confluence of 70-90%, they were serum-starved to induce differentiation.”

Comment #7

-In the introduction the authors referred to the beneficial effect of ASO as a treatment for the disease however the references shown are misleading, since they do not specifically address this issue. There are recent papers that were published showing this beneficial effect on skeletal muscle and the brain. Ait Benichou et al., 2021 and 2022.

Response: Using this information, we have altered the text to reflect this (references 28 and 29)  

Comment #8

-The authors should refer to the work in which ISIS 486178 was first described, used here as a control.

Response: Thanks to the reviewer for his/her comment, the information was added as a new reference 59.

Comment #9

-The authors should clearly state what was the concentration (or the volume%) of DMSO used in control treatments and what was the concentration of DMSO in 10microM concentration of the drugs, the highest used in the study. The data was normalized to DMSO treatments. The question is, did the DMSO has any effects? And at what concentration this was studied?

Response: We would like to thank the reviewer for his/her comments, accordingly, the DMSO final concentration has been properly addressed in the Materials and Methods section 4.3. “In this study, DMSO was used at a concentration of 0.05%, and no effects of DMSO were observed”.

Comment #10

Vorinostat reduced the DMPK mRNA levels (Figure 3), is this the mechanism of action of the drug?

Response: Treatment with vorinostat reduces DMPK mRNA level both in DM1 and control cells, in addition HSA mRNA is also decreased in vorinostat treated mice. We believe that there is a good likelihood that this is the mechanism of action of the drug.

Comment #11

-Figure 4A and Figure 4, in the supplementary data showing αMBNL1 expression the mouse where many lines are saturated and therefore unacceptable.

Response: Figure 4 was modified.

Comment #12

-The conclusion has to be revised since the disease is multisystemic. The authors did not show that the drug can be used to affect other organs such as the brain and the heart the most problematic aspect of the disease. Hence the conclusion and title need be to revised and reflect outcomes on skeletal muscle only.

Response: Using the reviewer’s comment, the title and the conclusion were modified. The title now state “Vorinostat Shows Improves Myotonic Dystrophy Type 1 Splicing Abnormalities in DM1 Muscle Cell Lines and Skeletal Muscle From a DM1 Mouse Model”and the conclusion state “Several DM1 disease markers were ameliorated by vorinostat in both in vitro and in vivo experiments. This indicates that vorinostat is a promising novel treatment for DM1 in muscle cells and skeletal muscle. There is a need for further investigation of the specific mechanism of action of vorinostat in DM1 in other organs, such as the brain and the heart.”

Round 2

Reviewer 2 Report

Icono de Validado por la comunidad Authors have answered correctly all of my questions.

Author Response

Dear Dr. Jidapha Techataratip

Assistant Editor,

Thank you for providing us the opportunity to submit a revised version of our manuscript entitled Vorinostat Improves Myotonic Dystrophy Type 1 Splicing Abnormalities in DM1 Muscle Cell Lines and Skeletal Muscle From a DM1 Mouse Model” to the International Journal of Molecular Sciences.

We thank the two reviewers for their thoughtful comments and suggestions regarding our manuscript. We have taken into account all of the comments and incorporated them into the revised manuscript. An itemized point-by-point response to the comments from reviewers is provided below in response to the changes made to the original document.

COMMENTS FROM REVIEWER 2

Comment #1

Authors have answered correctly all of my questions.

Response: Thanks for this, the authors would like to thank the reviewer for his/her comments during the revision process.

Reviewer 3 Report

The authors were only partially responsive to this reviewer's comments. Nevertheless, the authors must provide the data readily available and corresponding the compounds that reduce foci by 30 to 50%. These data can be provided as supplemental data if they cannot fit within the main text so that the readers are informed.

Author Response

Dear Dr. Jidapha Techataratip

Assistant Editor,

Thank you for providing us the opportunity to submit a revised version of our manuscript entitled Vorinostat Improves Myotonic Dystrophy Type 1 Splicing Abnormalities in DM1 Muscle Cell Lines and Skeletal Muscle From a DM1 Mouse Model” to the International Journal of Molecular Sciences.

We thank the two reviewers for their thoughtful comments and suggestions regarding our manuscript. We have taken into account all of the comments and incorporated them into the revised manuscript. An itemized point-by-point response to the comments from reviewers is provided below in response to the changes made to the original document.

COMMENTS FROM REVIEWER 3

Comment #1

The authors were only partially responsive to this reviewer's comments. Nevertheless, the authors must provide the data readily available and corresponding the compounds that reduce foci by 30 to 50%. These data can be provided as supplemental data if they cannot fit within the main text so that the readers are informed.

Response: Thank you for your comments regarding our manuscript. We have added information to the material and method section in response to the reviewer's comments (pages 15-16, lines 451-453), “The impact on foci area for all compounds screened along with change in nuclei count, serving as a proxy for cytotoxicity, can be found in supplementary Table S1.”, and (page 22, lines 777-778), “The rankings in Table S1 are based on the reduction in foci per nuclear area (green column).”, and in the results section (page 3, line 115), “The impact of all drugs screened on foci is shown in table S1.”. Based on the results of the compounds tested, Table S1 in the supplementary data provides ranked information about foci reduction.

Round 3

Reviewer 3 Report

No further comments.